# Assessment of the Levels of Oxidative Stress, Muscle Damage, and Psychomotor Abilities of Special Force Soldiers during Military Survival Training

**DOI:** 10.3390/ijerph17134886

**Published:** 2020-07-07

**Authors:** Paweł Różański, Ewa Jówko, Andrzej Tomczak

**Affiliations:** 1Department of Physical Education and Sport, Faculty of Physical Education and Health in Biała Podlaska, Józef Piłsudski University of Physical Education in Warsaw, 00-968 Warsaw, Poland; pawel.rozanski@awf-bp.edu.pl; 2Department of Natural Sciences, Faculty of Physical Education and Health in Biała Podlaska, Józef Piłsudski University of Physical Education in Warsaw, 00-968 Warsaw, Poland; 3Department of Education for Security, Faculty of National Security, War Studies University in Warsaw, 00-910 Warsaw, Poland; a.tomczak@akademia.mil.pl

**Keywords:** sleep deprivation, divided attention, forearm strength, prooxidant–antioxidant homeostasis, lipid peroxidation

## Abstract

The aim of this study was to analyze the changes in biochemical markers of oxidative stress and muscle damage, as well as psychomotor abilities during a military survival training. The study included 15 soldiers of special unit (SU), that completed 48 h military survival training combined with sleep deprivation. Before the training (P1), after 24 h (P2), and after 48 h of training (P3), blood samples were taken to measure biochemical markers. At the same time points, the measurements of divided attention and handgrip strength were conducted. Glutathione peroxidase activity decreased significantly at P3, in comparison with P1 and P2 (*p* < 0.0001), however, no changes were observed in other biochemical markers (i.e., lipid hydroperoxides, creatine kinase and superoxide dismutase activity) throughout the survival training (*p* > 0.05). The divided attention index was improved significantly at P2 and P3, as compared to P1 (*p* < 0.05). A tendency to change in maximum strength was found during the training period (main time effect; *p* = 0.08). Moreover, the strength differentiation (i.e., 50% maximum strength; 50%max) was higher at P3 than at P1 and P2 (*p* < 0.05). In conclusion, the 48 h survival training in the SU soldiers does not cause oxidative stress or muscle tissue damage, as well as any deterioration, and even improvement in psychomotor abilities. However, the change in strength differentiation (i.e., the production above 60%max instead of target 50%max) after the training may point to deterioration in motor control. Although it should be confirmed in further study with a more numerous group of soldiers, our findings indicate that the special unit soldiers will be able to perform, in a correct manner, specialized tasks related to their long-term activities, especially those which require divided attention. However, participation in long-term survival training, even with low workload, combined with sleep deprivation, results in a deterioration in motor control which may indicate the relevance of monitoring coordination motor abilities/skills in the training process of special unit soldiers.

## 1. Introduction

The security of Polish citizens is guaranteed by the constitution and is regulated by appropriate provisions and acts. This goal could be achieved through the creation of strong military structures. However, there may be situations when conventional armies may not be equipped to resolve conflicts [1]. Particularly, situations concerning terrorist attacks, hostage liberation, the capture of criminals, and counteracting organized crime groups are difficult [2]. Well trained special forces handle such interventions [3]. The Polish special forces constitute one of the five main components of the Polish armed forces. This special team has been created to resist the various threats of the modern world. Its strength lies in its efficiency and ability to take immediate action in challenging conditions [4]. 

At present, the special armed forces present in Poland comprise the following military units: GROM, FORMOZA, AGAT, NIL, and COMMANDO. Soldiers for each of these special force units around the world are carefully selected based on their performance in complex and rigorous training programs. Soldiers serving in such units should have superior psychophysical predispositions [5]. The competence of special force soldiers is directed at prolonged action in difficult circumstances combined with intense psychomotor load and sleep deficit. In addition to superior physical fitness, they need to possess certain characteristics such as general intelligence, the ability to learn and memorize, emotional maturity, strong personality, high social competence level, and the motivation for performing complex tasks. 

However, it has been pointed out that individuals with the ability to cope with difficult situations are the preferred candidates for these types of special forces [6]. Furthermore, they should be able to tolerate excessive physical effort in circumstances involving physical and psychological stress [6]. Due to the service’s specificity, the behavior displayed while performing actions in difficult and extreme conditions is analyzed during training [7]. Extreme psychophysical effort and associated stress produce specific biochemical, psychophysical, and defensive reactions which may be linked to the impairment of human motor functions. Of all the analyzed components (skills), special importance is assigned to psychomotor capabilities, especially the efficiency of cognitive processes, focused on the task, divided attention, speed of thought operations, deductive reasoning, and concluding. High proficiency in perception range, orientation, spatial imagination, and quick decision-making processes are all important traits that should characterize a special force soldier [1]. To prepare soldiers for service in special force units, real-world combat tasks connected with strong emotions are replaced by training simulations, in which the soldiers’ mental resilience is assessed through the efficiency of their actions performed under extreme conditions [8]. 

As mentioned above, survival training is characterized by a long duration (ranging from several hours to several dozen hours), with sleep deprivation and various psychophysical actions. It has been suggested that sleep deprivation may significantly reduce a person’s ability to perform tasks that require additional energy expenditure [9]. Moreover, as demonstrated by epidemiological and laboratory studies, sleep deprivation may decrease tolerance to exercise in extreme weather conditions, i.e., heat or cold [10].

Among other factors, reactive oxygen species (ROS) are presumed to accumulate during awakening, inter alia, due to enhanced metabolic activity, and are said to be responsible for the unfavorable effects of sleep deprivation [11,12]. During survival training, prolonged physical activity without an opportunity to sleep may contribute to unfavorable changes in prooxidant–antioxidant homeostasis with the resultant tissue injury. One potential consequence of these changes may be the deterioration of physical and mental performance. This finding is confirmed by the results of our previous study [13,14], in which young physically active men underwent 36 h survival training with physical activity at low intensity, combined with sleep deprivation, in the summer season. The survival-training-induced oxidative stress (exhibited by the impaired enzymatic antioxidant defense and increased lipid peroxidation in the blood) that, in turn, caused muscle damage (indicated by an increase in the blood creatine kinase (CK) activity). 

It is interesting to note that this deterioration in oxidant–antioxidant balance and muscle damage was found after 24 h of the training (after overnight activities) [13]. Similar findings were observed with regard to the deterioration in the ability to differentiate the use of forearm muscle strength after night training, whereas at the later stages of training (i.e., after 36 h), no significant change in this parameter was found [14]. In our recent study [15], a 36 h survival training program was conducted for the Polish Naval Academy cadets under cold ambient conditions (autumn). As indicated in the previous study [14], the ability to differentiate the strength of the forearm muscles deteriorated after the night part of the training (after 24 h). However, in the case of the Polish Naval Academy cadets, a gradual impairment in their ability to maintain balance was noted after 36 h of survival training that continued throughout the 12 h recovery period [15].

The training program conducted to prepare soldiers for military action aims at restricting their psychomotor functions and well-being under challenging and extreme environmental conditions [16,17]. Typically, facing such a situation at an early stage leads to reduced self-confidence, a limited control of problematic situations, and erroneous decision making concerning the completion of activities [2,18]. This may bear significance for special forces soldiers, who are forced to make a responsible decision regarding neutralizing the threat and eliminating the enemy in a short time (in stressful situations). The mentioned psychomotor reactions may be reflected by the quality of tasks performed by the member of the special forces [19], such as conducting covert navigation in a hostile terrain with limited visibility, orienteering march in a terrain patrolled by the enemy, and performing reconnaissance under conditions of fatigue, malnutrition, and sleep deficit.

Considering the above issues, the present study was undertaken to assess the level of biochemical markers (oxidative stress and muscle damage) in the blood, and divided attention along with forearm muscle (handgrip) strength indicators during military survival training for male special force unit soldiers.

## 2. Materials and Methods

### 2.1. Subjects

This study included 15 male special forces soldiers (age—33.1 ± 3.5 years; height—179.9 ± 6.4 cm; body weight—78.5 ± 5.2 kg). The soldiers had served in the special unit for 4–7 years and all had a high level of physical activity as confirmed by a physical activity questionnaire (IPAQ) that was completed as part of their baseline assessment. Due to the discreet nature of the training and the scope of actions performed by the Polish special forces described in the study, the present study does not mention its name or document the military training profile. However, it should be emphasized that the representatives of this formation are part of Poland’s leading special forces, conducting the highest number of special operations related to counterterrorism outside the country. The abbreviation SU (special unit) was used to characterize the study group. The soldiers voluntarily participated in the study and were debriefed before the trials were performed. The study was conducted in accordance with the principles of the Declaration of Helsinki. All the participants gave written consent to take part in the research, and the research protocol was approved by the Local Ethics Committee of the Józef Piłsudski University of Physical Education in Warsaw (No SKE 01-16/2014).

### 2.2. Study Design

The SU soldiers completed the preplanned, 48 h survival training program (included in their training schedule) targeted at performing covert activities, reaching designated places, and observing persons and objects, while remaining undercover all the time. The tasks of the soldiers included stealth movement in a specified terrain, under isolation, with sleep deprivation (the soldiers were not allowed to sleep for the first 24 h of the training, and during the subsequent 24 h of the training, the soldiers were allowed to sleep three hours in total). The possibility of being detected by patrols was considered as simulating activity under stress conditions. Throughout the training, they moved stealthily in the designated area and conducted observations while adapting to the conditions of the natural environment. The study was carried out between March and April, at temperatures between 4 °C at night and 11 °C during the day. The weather during the study was windless, with moderate cloud cover and no rainfall. 

The soldiers reported at specified times at checkpoints, stating the degree of task performance. Subsequently, they were provided with further instructions. Soldiers moved individually, paying attention to the imposed time regimen, place, and task. The motor load to which the SU soldiers were exposed (apart from the 15 kg equipment that had to be carried) was relatively low. Higher status was assigned to the psychological burden resulting from the time pressure to complete tasks in the encountered climatic and environmental conditions. Additionally, the SU soldiers executed their tasks near guarded areas, under the risk that their actions would be found out by search patrols. The tasks were performed as part of a military survival training, classified in military structures as SERE (survival, evasion, resistance, escape). 

Before joining the training and while performing the tasks, after reporting at the given checkpoint (i.e., after 24 and 48 h of training), the soldiers underwent an assessment of psychomotor functions (forearm strength measurement and a computer test for divided attention). During the training, the SU solders received meals in the form of military food rations (the daily energy intake amounted to 3600 kcal, with protein, fat, and carbohydrate contributing 18%, 26%, and 56% of the dietary energy intake, respectively). The soldiers covered an average distance of 6 km within 48 h of the field training. The SU soldiers spent most of their time undercover, constructed provisional shelters, concealed the traces of their presence, and conducted target observations. In the established time intervals, they were asked to reach reporting checkpoints and then return to the area of operation. 

Apart from the measurements conducted in the field of motor coordination capability, a blood sample (10 mL) was collected from each SU soldier on three occasions: before the training (phase 1; P1), after 24 h (phase 2; P2), and after 48 h of training (phase 3; P3). 

All the blood collections were completed in the morning: from 6 a.m. to 6.30 a.m. The blood collections before survival training were taken on an empty stomach (after rest at night and at least 10 h after the last meal), whereas the blood collections after 24 h and 48 h of the training were done at least 5–6 h after the last meal because during the training soldiers were provided with light meals, which were the same for everyone. They also had constant access to mineral water in unlimited quantities. 

### 2.3. Blood Sampling and Biochemical Analyses

Venous blood samples were drawn into heparinized test tubes and then centrifuged (for 10 min at 3000× *g* at a temperature of 4 °C) to separate erythrocytes and plasma. Subsequently, the erythrocytes were washed three times with a cold isotonic saline solution. Erythrocytes and plasma, as well as whole blood, were frozen and stored at −80 °C.

Measured parameters in the plasma included: creatine kinase (CK) activity—the index of muscle cell damage, and the concentration of lipid hydroperoxides (LOOHs)—the lipid peroxidation index. Superoxide dismutase (SOD) and glutathione peroxidase (GPx) were selected as representatives of the enzymatic antioxidant system. The activity of SOD was measured in red blood cells (after the centrifugation of whole blood and the washing of erythrocytes with the NaCl solution). In turn, the activity of GPx was assessed in whole blood.

The CK activity in the plasma was denoted by a kinetic method using a kit designed by the Alpha Diagnostics company (Warsaw, Poland). The diagnostic kits of the Randox company (Crumlin, UK) were used to measure the parameters of the enzymatic antioxidant system in the blood, i.e., the activity of SOD and GPx, that was expressed as units per gram of hemoglobin [U/gHb]. The concentration of hemoglobin was also measured using the Randox Diagnostic Kit. The concentration of LOOHs in the plasma was marked as described previously [13,20].

### 2.4. Divided Attention and Handgrip Strength

The measurements of divided attention and handgrip strength were conducted at the specified intervals, identical to the intervals between the collection of venous blood collection for biochemical tests. Reactions in the field of divided attention were tested using the Computer Coordination Capabilities Test software [21], whereas the handgrip strength measurement was performed with a mechanical hand dynamometer (PZA/3359 dynamometer—Fabrication Enterprises Inc., Westchester County, NY, USA).

Previous studies have described the procedure of the divided attention computer test in detail [14,15]. The measurement of divided attention was performed as follows [21]. The tested person sat in front of the monitor, put their fingers on the keyboard, and then the person who carried out the test initiated it by pressing the button. The laptop screen (15.6”) displayed two types of signals:

The first type of signals was figures in the center of the screen: a square, a circle, and a cross. If they were displayed in the right order (i.e., the square, the circle, and then the cross), the person had to press “+” with the thumb of the right hand (or the “Q” key with the thumb of the left hand) when the cross appeared. Every other sequence of figures was incorrect.

The second type of signals was small squares displayed in the corners of the laptop screen. If one of the corners of the screen displayed four small squares at the same time, the person had to press “-” with the forefinger of the right hand (or the “1” key with the forefinger of the left hand). The better the result was, the more valid signals were observed and “received”, while the number of errors (missed signals or incorrect key presses) negatively impacted the test result. At the end of the test, the following results were presented: the number of perceived signals, the number of errors (omitted signals and incorrectly pressed keys), and the respective percent indices (i.e., the percentage of correct answers in relation to the total number of signals). The test lasted about 90 s (it ended automatically). The reliability coefficient for the aforementioned test is r = 0.92 [21].

Before the first test (P1) was conducted, the SU soldiers were familiarized with its principles in the abovementioned software and a trial test was performed (to eliminate the effect of learning in the subsequent tests, i.e., P2 and P3). Furthermore, the randomness of the signals appearing on the computer screen, as well as their programmed variability, prevented test learning and the memorization of the order of signals. Tests on the attention index assessment were carried out individually, in a separate and quiet room, without the presence of third parties. At each phase (P1, P2, and P3), the test was only performed once, without the possibility of its repetition. Before each divided attention test, the participant was reminded of the principles of its performance. 

The handgrip strength test (provided in N) was conducted in a standing position, on the dominant limb, as described in the previous work [15]. Before the measurement, the SU soldiers were familiarized with the operation of the dynamometer, followed by 3 subsequent tasks. Each of the soldiers was asked to perform 3 tasks to measure their force differentiation skill. The maximum grip of the dynamometer band was recorded during the first task (max). The second task involved the differentiation of the force by the soldier in such a manner that he performed the dynamometer grip attaining half of the value obtained in the first task (50%max). Before the third task, the investigator provided the examined soldier with information on the error made in the second task. The information was provided to enable the examined soldier to perform a correction of the force used in the preceding task (corrected 50%max). Every task was repeated five times and averaged [22].

### 2.5. Statistical Analyses

Biochemical parameters and divided attention were analyzed using a one-way ANOVA with the Bonferroni post-hoc test for multiple comparisons. The normal distribution of all variables was confirmed with the Shapiro–Wilk test and visual inspection (quantile distribution plots). The Statistica v. 10.0 software package (StatSoft, Krakow, Poland) was used for the calculations. The values were presented as the means with standard deviation (SD). The level of statistical significance was set at *p* < 0.05.

## 3. Results

The 48 h survival training did not significantly affect the plasma CK activity (Table 1). Likewise, the concentration of plasma LOOHs and SOD activity in erythrocytes (Table 1) did not change significantly during the training. On the contrary, the training in the examined group of soldiers caused significant changes in the GPx activity in whole blood (with the main time effect, *p* < 0.001). After 48 h of survival training, a significant decrease in GPx activity was observed in comparison to the baseline value and the value after 24 h of training.

Table 2 presents the mean values of the divided attention and handgrip strength indices during survival training. Based on the obtained results included in Table 2, it can be observed that the tested divided attention index in the subsequent training phases exhibited an increasing trend (with a main time effect; *p* < 0.05) despite the limited amount of rest (lack of sleep). At the commencement of the study (P1), the mean values of the divided attention attained 43% of maximum capacity. After 24 h of the training (P2), its value increased by an average of 9% (*p* < 0.05 as compared to P1), and after 48 h of survival activity without the possibility of sleep the tested parameter increased by another 2% (*p* < 0.05 as compared to P1). 

The maximum handgrip strength (Table 2) showed a tendency to change significantly after the 48 h training period (main time effect; *p* = 0.08). For strength differentiation (50%max), a significant increase in this index was observed after 48 h of training relative to the baseline levels and values after 24 h of training (with main time effect; *p* < 0.05). 

## 4. Discussion

In the present study carried out with soldiers of the special unit, the 48 h survival training combined with sleep deprivation did not cause oxidative stress or muscle tissue damage. Moreover, the soldiers did not show any deterioration in psychomotor abilities. On the contrary, there was a slight improvement in the divided attention index.

It is believed that physical stress, such as those related to prolonged and chronic exercise, promotes the generation of ROS, which, in turn, leads to exercise-induced muscle injury [23]. One of the harmful effects of ROS overproduction occurring in oxidative stress conditions is lipid peroxidation, which may be indicated by an increase in the blood levels of LOOHs [24]. The peroxidation of membrane lipids is one of the factors which may cause a loss of muscle cell membrane fluidity and an increase in membrane permeability and thereby lead to an exercise-induced release of muscle enzymes to blood [25].

In our previous study on physical education students [13], long-term physical activity during 36 h survival training (even at low intensity) along with a lack of sleep induced oxidative stress and caused damage to muscle tissues. The significant positive correlation between plasma CK activity and plasma LOOHs provides evidence for the link between free radicals and muscle injury [13]. Keeping these points in mind, in the current study, the lack of changes in the LOOHs level in response to the 48 h training can explain the absence of CK response to the workloads during the training of SU soldiers. 

It must be emphasized that ROS are neutralized by antioxidant systems, including the enzymatic system (e.g., superoxide dismutase, glutathione peroxidase) [26]. In the present study, an initial low level of both LOOHs and CK in plasma was found in SU soldiers, along with the lack of changes in these parameters in response to the 48 h training, which can be explained by the high levels of antioxidant defense at the baseline. Pretraining GPx activity in the blood seems to be higher in SU soldiers (the present study) in comparison with students [13]. Despite the differences mentioned above, in SU soldiers (the current study) and civilians [13], the survival training with sleep deprivation resulted in a reduced activity of enzymatic antioxidant defense. However, it must be emphasized that changes in all the parameters of oxidant–antioxidant homeostasis should be taken into account while interpreting the results. 

To be more precise, in our previous study [13], a decrease in SOD and GPx activity in the students after 36 h survival training was preceded by an increase in LOOHs and CK activity after 24 h of the training. Therefore, the impairment of the antioxidant system observed in the group of physical education students exposed to survival training [13] might be associated with the partial inactivation of enzymatic proteins, resulting from their allosteric or covalent modifications induced by ROS [27]. Moreover, the consequence of that deterioration in the antioxidant system and oxidative stress was a significant decrease in the performance capabilities of the students [14]. Moreover, in our latest study carried on Naval cadets, the 36 h military survival training resulted in the deterioration of the selected coordination of motor abilities [15], and those changes were accompanied by an intensification of the oxidative stress [data in publication process]. On the contrary, in the present study, taking into account low levels of oxidative stress and muscle damage parameters throughout the survival training, it is most likely that the reason for the post-training decrease in GPx activity might be a decrease in ROS, especially hydrogen peroxide, which is a substrate for this enzyme. According to some authors [28], GPx may protect against oxidative stress, but, in excess, it may also have deleterious effects (in the form of diminished mitochondrial function and decreased cellular metabolism) due to the absence of essential cellular oxidants playing an important role as cellular signal transducers.

Summing up, the results of our previous studies (as explained above) suggest that a deterioration in physical and mental performance during survival training without an opportunity to sleep may occur, at least partially, due to the disturbance of oxidant–antioxidant balance toward oxidative stress. All those adverse changes were probably the response to the exposure to low physical demands, which is related to the specificity of the military survival training [29], as well as psychological stressors related to, inter alia, sleep deprivation. 

The association between sleep deprivation (itself) and oxidative stress is well documented in animal studies [30,31], possibly as a result of an increase in energy expenditure [9]. Additionally, taking into account the results of animal studies, the consequences of sleep deprivation associated with mild to moderate multi-organ damage are at least partially mediated by inflammation and reactive oxygen species [32]. 

The reverse is true in the case of the soldiers in the current study. Survival training in SU soldiers, despite the workload (comparable to that of physical education students or naval cadets) and lack of sleep, did not exacerbate oxidative stress or cellular damage. Moreover, the maximal handgrip strength did not change significantly throughout the 48 h of survival, whereas the ability to differentiate the strength of the forearm muscles increased after 48 h of training (as compared to the level at baseline and after 24 h of training). In addition, the analyzed index of divided attention increased after both 24 and 48 h of the survival training (as compared to the baseline). It may suggest a high psychomotor resistance of the soldiers during their tasks, and thus a high ability to adapt to stressful conditions. This phenomenon was described earlier when it was suggested that a psychological burden may affect the performance of tasks in the following way: as physiological arousal increases, it is easier to perform an activity/task, but only to a certain level. When this level is exceeded, performance decreases, leading to the extreme disintegration of behavior in emergencies [14]. On the other hand, while the improvement in the divided attention index throughout the survival training indicates the mobilization of some psychomotor functions in a stressful situation, the increase in strength differentiation in this time should be interpreted with caution. Namely, greater force production during the 50% max target test at P3 (i.e., above 60%max) may indicate that the soldiers were unable to target 50% of their initial maximum. This inability to gauge their effort rather points to the deterioration of motor control, probably due to the negative influence of sleep deprivation [33]. However, as reported in a study carried out on military officers [34], 60 h sleep deprivation itself did not affect the maximal muscle strength and maximal rate of force development of the knee extensors, as well as motor control (target movement of the right hand) and balance. In turn, in another study [35], compared to men who have been deprived of sleep alone, those performing 5 h of intermittent moderate exercise during 30 h of sleep deprivation appeared to be more vulnerable to impairment in both cognitive and psychomotor performance (two-choice reaction times). Therefore, taking into account the results of the current and aforementioned studies, it seems that physical exercise during sleep deprivation may intensify the deterioration of some parameters of psychomotor performance. It can be confirmed by our previous study on Polish Naval Academy cadets [15], in which a 36 h survival training with sleep deprivation caused impairments in selected coordination motor abilities, like the ability to differentiate the strength of the forearm muscles (after the night part of the training) and the ability to maintain balance (throughout the training period). 

However, to confirm the cumulative stress experienced during the military training, stress hormones response to the training should have been measured, which is the limitation of this study. On the other hand, other investigators have reported hormonal changes after the military training [29,36]. Szivak et al. [36] measured a neuroendocrine response to the military SERE training in men serving in the U.S. Navy and Marine Corp. They found an increase in stress hormone concentrations (epinephrine, cortisol), with a concomitant reduction in testosterone concentrations, but these parameters were assessed after as long as after 10 days of training [36]. Moreover, hormonal responses to the real-world stress associated with the military training (with modest physical demands) were investigated in male SERE school students [29]. The study revealed an increased release of catabolic hormones and suppressed anabolic hormone release after the captivity and interrogation segments of SERE (lasting 1 week) [29]. Therefore, further study is needed in SU soldiers with the assessment of endocrine response to 48 h military survival training. 

## 5. Conclusions

The 48 h survival training carried out in special unit soldiers did not cause oxidative stress or muscle tissue damage. Moreover, the soldiers did not show any deterioration, but instead showed improvement in the divided attention index. Based on the analyzed biochemical and psychomotor indicators, it can be stated that the special unit soldiers should be able to carry out their specialized tasks, especially those related to divided attention, at an optimum level for long periods of time. However, participation in long-term survival training, even with a low workload, combined with sleep deprivation, results in a deterioration in motor control which may indicate the relevance of monitoring coordination motor abilities/skills in the training process of special unit soldiers.

## Figures and Tables

**Table 1 ijerph-17-04886-t001:** Blood levels of the biochemical parameters in the soldiers throughout 48 h of military survival training (*n* = 15).

Survival Training Phase	CK Activity [U/L]	LOOHs [µmol/L]	SOD Activity [U/gHb]	GPx Activity [U/gHb]
P1	184.7 ± 66.1 ^a^	0.68 ± 0.26 ^a^	1196.9 ± 126.5 ^a^	62.5 ± 15.9 ^a^
P2	145.6 ± 38.5 ^a^	1.00 ± 0.97 ^a^	1178.6 ± 191.9 ^a^	54.6 ± 15.7 ^a^
P3	151.7 ± 58.3 ^a^	0.73 ± 0.38 ^a^	1201.0 ± 157.2 ^a^	36.4 ± 13.1 ^b^
Main time effect	*p* = 0.13	*p* = 0.33	*p* = 0.91	*p* < 0.0001

Values are mean ± SD. ^a,b^ differences in values of the analyzed parameters in the group (One-way ANOVA). Values (at different time points) that do not have a common letter differ significantly (*p* < 0.05). P1—phase 1 (before survival training); P2—phase 2 (after 24 h of survival training); P3—phase 3 (after 48 h of survival training); CK—creatine kinase; LOOHs—lipid hydroperoxides; SOD—superoxide dismutase; GPx—glutathione peroxidase; CK and LOOHs were measured in plasma; SOD activity was determined in erythrocytes; GPx activity was analyzed in whole blood; activities of antioxidant enzymes in erythrocytes and whole blood are expressed in U per gram of hemoglobin.

**Table 2 ijerph-17-04886-t002:** Mean values (±SD) of divided attention and handgrip strength in special forces soldiers (*n* = 15) during the 48-h military survival training.

Survival Training Phase	Divided Attention (%)	Handgrip Strength [N]			
		Max	Max/2	50%max	Corrected 50%max
P1	43 ± 18 ^a^	392.4 ± 49.2	196.2 ± 24.6	220.7 ± 38.5 ^a^	206.3 ± 34.1
P2	52 ± 20 ^b^	367.2 ±56.9	183.5 ± 28.4	215.0 ± 35.7 ^a^	198.5 ± 33.9
P3	54 ± 24 ^b^	410.1 ± 44.4	205.0 ± 22.2	254.0 ± 48.5 ^b^	218.4 ± 32.7
Main time effect	*p* < 0.05	*p* = 0.08	*p* = 0.07	*p* < 0.05	*p* = 0.27

^a,b^ Differences in values of the analyzed parameters in the group (One-way ANOVA). Values (at different time points) without a common letter differed significantly (*p* < 0.05). P1—phase 1 (before survival training); P2—phase 2 (after 24 h of survival training); P3—phase 3 (after 48 h of survival training).

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
