# Peer review of "Assessment of the Levels of Oxidative Stress, Muscle Damage, and Psychomotor Abilities of Special Force Soldiers during Military Survival Training"

_ijerph, 2020, doi:10.3390/ijerph17134886_

Round 1
Reviewer 1 Report
Dear Authors,
Thank you for your interesting manuscript. Before I can recommend its publication, please carefully assess the following issues:
Major issues:
- Your key term "attention divisibility" is not easily understood in English, and presumably a calque from Polish. In fact, "attention divisibility" may be interpreted as an impairment of attention, not as a favourable ability. (cf. e. g. https://forum.wordreference.com/threads/divisibility-of-attention.2031185/) Please rephrase throughout the paper so that its meaning is clear to Standard English language readers.
- You focus on aspects of (conceivable) activity-induced muscle damage due to a disturbance of the oxidant-antioxidant balance towards oxidative stress in your study. However, at the same time, you state that the SU soldiers under study were subject to "a relatively low" motor load throughout the 48 h training. Hence, I must question the validity of your study design in that respect: How can you adequately assess muscle damage (and/or its possible causes) under “low” motor load conditions?
- Apart from the biochemical markers you have investigated, it is conceivable that a severe endocrine response takes place during and/or after the 48 h training, especially in terms of increased cortisol and decreased testosterone levels (in case of male subjects) and changed adrenaline levels. Why have you not investigated those important hormones? Please at least discuss on that reasonably. It is a clear weakness of your paper.
Minor issues:
- lines 19, 21: Font size
- line 50: …”pointed *out*…”
- lines: 50–52: Role of neuroticism is unclear; no references are given. Please improve.
- line 108: Please provide gender of subjects – all male?
- line 117: Please refer to the Declaration of Helsinki, if applicable
- line 126: Please be more specific than “relatively low”, e. g., provide MET values
- line 128: Be more specifc on climatic and environment conditions at this point (already). What were temperature, humidity, solar stress (if applicable) etc. – Move information from lines 143-144 here.
- line 137: Was it 3600 kcal in total over 48h or divided in what portions? Please provide carbohydrate, protein and fat content as their ratios may substantially affect systemic metabolic response
- lines 159–162 Font size.
Best regards
Author Response
We are extremely grateful for all critical remarks concerning our work indicating your very profound analysis of our manuscript. We have attached a point-by-point responses to the reviewer's comments.
Response to Reviewer 1 Comments
We are extremely grateful for all critical remarks concerning our work indicating your very profound analysis of our manuscript.
Major issues:
Point 1. Your key term "attention divisibility" is not easily understood in English, and presumably a calque from Polish. In fact, "attention divisibility" may be interpreted as an impairment of attention, not as a favourable ability. (cf. e. g. https://forum.wordreference.com/threads/divisibility-of-attention.2031185/) Please rephrase throughout the paper so that its meaning is clear to Standard English language readers.
Response 1: Thank you for this remark. According to your suggestion, we have changed "attention divisibility" to “divided attention” (the term used in other papers, references below) throughout our manuscript.
References:
- Bondallaz P, Chtioui H, Favrat, E.Fornari, C.Giroud, P.Maeder. Assessment of Cannabis Acute Effects on Driving Skills: Laboratory, Simulator, and On-Road Studies; in Handbook of Cannabis and Related Pathologies, Elsevier; 2017, Pages 379-390 (Divided attention task- neuropsychological tests related to the ability of a subject to perform two different tasks simultaneously).
2. Moisala M, Salmela V, Salo E, et al. Brain activity during divided and selective attention to auditory and visual sentence comprehension tasks. Front Hum Neurosci. 2015;9:86. Published 2015 Feb 19. doi:10.3389/fnhum.2015.00086
3. Rill RA, Farago ́KB, Lőrincz A (2018) Strategic predictors of performance in a divided attention task. PLoS ONE 13(4):e0195131. https://doi.org/10.1371/journal.pone.0195131
- Verghese J, Buschke H, Viola L, et al. Validity of divided attention tasks in predicting falls in older individuals: a preliminary study. J Am Geriatr Soc. 2002;50(9):1572‐1576. doi:10.1046/j.1532-5415.2002.50415.x
- Hahn B, Wolkenberg FA, Ross TJ, et al. Divided versus selective attention: evidence for common processing mechanisms. Brain Res. 2008;1215:137‐146. doi:10.1016/j.brainres.2008.03.058
Point 2. You focus on aspects of (conceivable) activity-induced muscle damage due to a disturbance of the oxidant-antioxidant balance towards oxidative stress in your study. However, at the same time, you state that the SU soldiers under study were subject to "a relatively low" motor load throughout the 48 h training. Hence, I must question the validity of your study design in that respect: How can you adequately assess muscle damage (and/or its possible causes) under “low” motor load conditions?
Response 2: As described in the manuscript, the results of our previous studies in civilians (lines 287- 291) and Naval cadets (lines 311-314) indicate that long-term physical activity during military survival training (even at low intensity) along with lack of sleep induced oxidative stress and caused damage to muscle tissues, as confirmed by the increase in the activity of creatine kinase in plasma. Thus, our previous findings may indicate that not only physical but also psychological stress, related among other to sleep deprivation was responsible to oxidative stress and its consequences. This issue has been widely described in our previous paper (Jówko et al., 2018; also lines 76-87 of our current manuscript). Also, as described in the Introduction, sleep deprivation may reduce the ability to perform tasks that require additional energy expenditure and decrease tolerance to exercise (even at low intensity) in extreme weather conditions, i.e., heat or cold (references 9-10 cited in our manuscript; lines 72-75).
It has been suggested that free radicals are accumulated during waking, among other as a result of enhanced metabolic activity and they are responsible for the effects of sleep deprivation [Özdemir et al., 2013]. Sleep deprivation in rats resulted in an inability to retain body heat, an increase in energy expenditure. The metabolism of glycogen and other sources of stored energy results in the generation of oxidants or free radicals [Engle-Friedman, 2014].
The association between sleep deprivation (itself) and oxidative stress is well documented in animal studies [McEwen, 2006; Villafuerte et al., 2015]. It has been suggested that beside central nervous system, oxidative stress may regard other organs, like lung, heart, liver and skeletal muscle [Villafuerte et al., 2015]. In this line, increases in plasma aminotransferases: aspartate (AST) and alanine (ALT) was observed, without a change in their ratio, that suggest involvement of tissues besides the liver, such as muscle and possibly the heart [Everson et al., 2005]. Taking into account the results of animal studies, the consequences of sleep deprivation associated with mild to moderate multi-organ damage are at least partially mediated by inflammation and reactive oxygen species [Periasamy et al., 2015]. These findings confirmed the results of the earlier human study [Ilan et al., 1992], in which sleep deprived (72 h) young male volunteers reported increased plasma AST and ALT level.
Taking together, we have included some issues (mentioned above) in the text of revised manuscript (lines 326-327; 328-332).
References:
- Özdemir PG, Selvi Y, Özkol H, Aydın A, Tülüce Y, Boysan M, Beşiroğlu L: The influence of shift work on cognitive functions and oxidative stress. Psychiatry Res 2013; 210(3): 1219-25.
- Everson CA, Laatsch CD, Hogg N: Antioxidant defense responses to sleep loss and sleep recovery. Am J Physiol Regul Integr Comp Physiol 2005; 288(2): 374-83.
- Engle-Friedman M: The effects of sleep loss on capacity and effort. Sleep Sci 2014; 7(4): 213-24.
- McEwen BS: Sleep deprivation as a neurobiologic and physiologic stressor: allostasis and allostatic load. Metab Clin Exp 2006; 55: 20-3.
- Periasamy S, Hsu DZ, Fu YH, Liu MY: Sleep deprivation-induced multi-organ injury: role of oxidative stress and inflammation. EXCLI J 2015; 14: 672–83.
- Ilan Y, Martinowitz G, Abramsky O, Glazer G, and Lavie P: Prolonged sleep- deprivation induced disturbed liver functions serum lipid levels, and hyperphosphatemia. Eur J Clin Invest 1992; 22: 740–43.
- Villafuerte G, Miguel-Puga A, Murillo Rodríguez E, Machado S, Manjarrez E, and Arias-Carrión O: Sleep Deprivation and Oxidative Stress in Animal Models: A Systematic Review. Oxid Med Cell Longev 2015; ID 234952, http://dx.doi.org/10.1155/2015/234952
Point 3. Apart from the biochemical markers you have investigated, it is conceivable that a severe endocrine response takes place during and/or after the 48 h training, especially in terms of increased cortisol and decreased testosterone levels (in case of male subjects) and changed adrenaline levels. Why have you not investigated those important hormones? Please at least discuss on that reasonably. It is a clear weakness of your paper.
Response 3: We completely agree with your opinion. According to your comment, it has been added as the limitation of the study in the revised manuscript (lines 353-355). However, in contrast to oxidative stress indicators, endocrine response to the military training has already been addressed by many Authors (Morgan et al., 2000; Liebermanin et al., 2016; Szivak et al., 2018). We have described the results of other Authors related to endocrine response to the military training (lines 356-365).
References
- Lieberman HR, Farina EK, Caldwell J, Williams KW, Thompson LA, Niro PJ, Grohmann KA, McClung JP. Cognitive function, stress hormones, heart rate and nutritional status during simulated captivity in military survival training. Physiol Behav. 2016;165:86‐97. doi:10.1016/j.physbeh.2016.06.037
- Szivak, T.K., Lee, E.C., Saenz, C., Flanagan, D., Focht, B.C., Volek, J.S., Maresh, C.M., Kraemer, W.J. Adrenal Stress and Physical Performance During Military Survival Training. Aerosp. Med. Hum. Perform. 2018, 89(2), 99‐107.
- Morgan CA 3rd, Wang S, Mason J, et al. Hormone profiles in humans experiencing military survival training. Biol Psychiatry. 2000;47(10):891‐901. doi:10.1016/s0006-3223(99)00307-8
Minor issues:
- lines 19, 21: Font size
- line 50: …”pointed *out*…” (line 53)
- lines: 50–52: Role of neuroticism is unclear; no references are given. Please improve.
Response: Indeed, this term is confusing, what did not our intention (translation process) To avoid misunderstanding, it has been removed (lines 53-54).
- line 108: Please provide gender of subjects – all male? (lines 114,117)
- line 117: Please refer to the Declaration of Helsinki, if applicable (lines 127-128)
- line 126: Please be more specific than “relatively low”, e. g., provide MET values
Response: As described in the Methods, our soldiers covered an average distance of 6 km within 48 hours of performing tasks in the field (lines 163-164) with the 15-kg equipment that had to be carried (lines 147-148). However we were not able to control exercise intensity due to the specificity of the military survival training of our participants (special unit soldiers). To investigate the magnitude of stress soldiers may experience both in a training environment and operationally, it is helpful to study the effects of military training in as realistic of a setting as possible. Thus, due to the classified nature of certain aspects of the military training, the exact description and control of training loads and intensity often is not possible. Please, see the response 4 (below) to the comment of reviewer 2.
line 128: Be more specifc on climatic and environment conditions at this point (already). What were temperature, humidity, solar stress (if applicable) etc. – Move information from lines 143-144 here. (lines 139-143)
- line 137: Was it 3600 kcal in total over 48h or divided in what portions? Please provide carbohydrate, protein and fat content as their ratios may substantially affect systemic metabolic response (lines 159-161)
- lines 159–162 Font size.
Reviewer 2 Report
Rozanski and colleagues assessed changes in plasma indicators of oxidative stress and psychomotor abilities of Polish Special Unit soldiers over the course of a 48-hour military survival training exercise. Though glutathione peroxidase was reduced following the training exercise, biochemical markers of muscle damage (e.g., creatine kinase) were unchanged throughout. While the authors state that measures of psychomotor performance (attention divisibility and grip strength) improved, as presented I found these findings to be unclear. Overall, I found the study to be scientifically sound and worthy of consideration, however I ask that the authors please address the following concerns:
- The manuscript would benefit from English editing services. For example, in line 50 of the Introduction the authors describe preferred candidates for special forces as being “ambivalent”, meaning “having mixed feelings”. I do not believe this is the word the authors were looking for here and this is only one of many places in the manuscript where such grammar and syntax errors were made.
- The conclusion of the abstract is a bit presumptuous given the size of this study. Please reword to avoid conjecture.
- Would the authors be able to provide any greater detail regarding the participants? Specific characteristics of interest would be the duration of which they had served in the special forces as well as their typical exercise histories.
- I found the training protocol to be a bit vague and nondescript. Would the authors be able to provide any quantitative metrics such as number of steps taken per day or hours of sleep restriction?
- Given its importance in the current paper, greater detail regarding the assessment of attention divisibility is required. What types of questions were employed? How was it administered? How long did the assessment take? How was it scored?
- Collection, reporting, and interpretation of grip strength data is also a bit unclear. It is stated that each participant completed the task five times? This seems excessive as grip strength assessment is almost never administered more than twice. Can you provide the rationale for doing so? Though I have never see it before, I found the submaximal grip strength assessment in which the subjects were asked to target 50% of their initial maximum to be quite interesting. Curiously, it seems that during P3 the subjects consistently produced more force than the target, which the authors seems to interpret as being good. While incontestably the soldiers produced more force during P3, in this context I do not agree with the notion that this is beneficial. Rather, the greater force production during the 50% max target test, in the absence of a significant improvement in maximal strength, is evidence that the soldiers were unable to gauge their effort as well as they were before the training exercise. As a result, when asked to produce 50% of maximal force ~60% max force was produced, which is in fact evidence of reduced psychomotor performance (i.e., motor control). As this data is a central component of the paper, substantial revisions are required.
Author Response
We would like to express our gratitude for all critical comments concerning our manuscript.
We have attached a point-by-point responses to the reviewer's comments.
Response to Reviewer 2 comments
We would like to express our gratitude for all critical comments concerning our manuscript.
Point 1. The manuscript would benefit from English editing services. For example, in line 50 of the Introduction the authors describe preferred candidates for special forces as being “ambivalent”, meaning “having mixed feelings”. I do not believe this is the word the authors were looking for here and this is only one of many places in the manuscript where such grammar and syntax errors were made.
Response 1: Thank you for this comment. The manuscript has been re-analyzed by the translation service. We received a certificate confirming that the text is grammatically correct.
Point 2. The conclusion of the abstract is a bit presumptuous given the size of this study. Please reword to avoid conjecture.
Response 2: We agree with the reviewer, that our finding should be confirmed in further study with more numerous group of soldiers (it has been added to the conclusions in the abstract, lines 26-27). Also, our conclusions have been modified in a more cautious manner (lines 28-30; 367-375).
Point 3. Would the authors be able to provide any greater detail regarding the participants? Specific characteristics of interest would be the duration of which they had served in the special forces as well as their typical exercise histories.
Response 3: Some details have been added in revised manuscript (lines 118-120). We cannot give other details about the participants due to the discreet nature of the training and the scope of actions performed by the Polish special forces, as described in the manuscript (lines 121-125).
Point 4. I found the training protocol to be a bit vague and nondescript. Would the authors be able to provide any quantitative metrics such as number of steps taken per day or hours of sleep restriction?
Response 4: As described in the Methods, the motor load to which the SU soldiers were exposed was relatively low, since our soldiers covered an average distance of 6 km within 48 hours of performing tasks in the field (lines 163-164) with the 15-kg equipment that had to be carried (lines 147-148). However we are not able to provide any quantitative metrics.
To investigate the magnitude of stress soldiers may experience both in a training environment and operationally, it is helpful to study the effects of military training in as realistic of a setting as possible. Thus, due to the classified nature of certain aspects of the military training, the exact description and control of training loads often is not possible. The general purpose of SERE (Survival, Evasion, Resistance, Escape) is to teach soldiers how to survive in austere environments, how to evade capture by the enemy, how to resist exploitation and survive if captured, and ultimately how to escape capture (Szivak et al., 2018). During SERE, soldiers are exposed to a multitude of stressors which may be faced in a survival or captivity situation: environmental extremes (i.e., heat or cold exposure), physical demands, sleep deprivation, and psychological stress (Szivak et al., 2018).
As reported by Liebermanin et al. (2016), the acute psychological stress, with modest physical demands, associated with real-world SERE training: 1) substantially degraded mental and psychological functioning (based on multiple cognitive tasks and mood scales) 2) activated the HPA system, as demonstrated by increased levels of the catabolic hormone cortisol and suppressed release of the anabolic hormone testosterone; and 3) induced sympathetic nervous system arousal as indicated by substantially elevated epinephrine, norepinephrine, and heart rate in the absence of highly strenuous physical activity.
Likewise in study of Liebermanin et al. ( 2016), in our study, higher attention was paid to psychologically stressful activities, related to performing covert activities, reaching designated places under the time pressure, observing persons and objects, while remaining undercover all the time, with the risk of being detected by searching patrols. Tasks of the soldiers included stealth movement in a specified terrain, performing under isolation conditions, with sleep deprivation. The soldiers were not allowed to sleep for the first 24 hours of the training, and during the subsequent 24 hours of the training, the soldiers were allowed to sleep three hours in total.
Except for sleep deprivation, most of issues mentioned above related to our survival training are described in Methods (we only have added some information: lines 148-151). Also, we have added the information about hours of sleep restriction to revised manuscript (lines 136-138).
Point 5. Given its importance in the current paper, greater detail regarding the assessment of attention divisibility is required. What types of questions were employed? How was it administered? How long did the assessment take? How was it scored?
Response 5: In the present study, we did not describe the test for divided attention, as the exact measurement procedure was described in our previous works (Tomczak et al., 2017; Tomczak et al., 2019). However, taking into account the reviewer's comment, we have completed our previous description (lines 205-222).
Point 6. Collection, reporting, and interpretation of grip strength data is also a bit unclear. It is stated that each participant completed the task five times? This seems excessive as grip strength assessment is almost never administered more than twice. Can you provide the rationale for doing so? Though I have never see it before, I found the submaximal grip strength assessment in which the subjects were asked to target 50% of their initial maximum to be quite interesting. Curiously, it seems that during P3 the subjects consistently produced more force than the target, which the authors seems to interpret as being good. While incontestably the soldiers produced more force during P3, in this context I do not agree with the notion that this is beneficial. Rather, the greater force production during the 50% max target test, in the absence of a significant improvement in maximal strength, is evidence that the soldiers were unable to gauge their effort as well as they were before the training exercise. As a result, when asked to produce 50% of maximal force ~60% max force was produced, which is in fact evidence of reduced psychomotor performance (i.e., motor control). As this data is a central component of the paper, substantial revisions are required.
Response 6: In fact, In another study (Szivak et al, 2018), three attempts were performed in the test for grip strength using handgrip dynamometer. However, in our current work, we have implemented a force measurement procedure (described and used in our previous studies: Tomczak et al., 2017; Tomczak et al., 2019), which was developed by Polish specialists in the assessment of coordination skills and assumed five repetitions of the task (Juras, G., Waśkiewicz, Z. Temporal, spatial, and dynamic aspects of coordination motor abilities. [in Polish] AWF Katowice, 1998). We added this item to the references in our revised manuscript [22].
In our current study, the changes in 50% of maximum strength was proportional to the change in maximum handgrip strength. Since in the case of the maximum strength, only a tendency to a significant time effect was shown (p=0.08; table 2), we did not show post-hoc results (according to reviewers' comments in our previous works), despite a significant increase in the maximum strength after 48 h compared to the values after 24 h (p= 0.025). Moreover, we have reanalyzed all max strength and 50% max strength results, and we have found a significant positive correlation between the variables (r= 0.63; p= 0.000004). Also, positive correlations have been found between % changes in max strength and % changes in 50% max strength (P1-P2; P1-P3, P2-P3; with r values above 0.60; p< 0.0001).
Our previous study confirm this finding (Tomczak et al., 2017). In that study, during the entire training, the maximum force of the forearm muscles remained at a similar level. However, it was reduced after 12 hours of recovery period (as compared to pretraining value, as well as the values at 24 and 36 hours of survival training). A similar relationship was observed for 50% of maximal force (it decreased after 12-hour recovery period, as compared to pretraining).
Taking together, to avoid confusion, we have decided to place post-hoc analysis in max strength (table 2) and complete the description in the text (lines 22-23; 267-270).
Round 2
Reviewer 1 Report
Dear Authors,
Thank you very much for your careful revision and detailed responses to my items. You have successfully addressed all of my concerns.
It is now a well-written, interesting manuscript that merits publication in IJERPH.
Best regards
OUe
Author Response
We would like to express our gratitude for all critical comments concerning our manuscript.
Reviewer 2 Report
I would like to thank the authors for considering my comments and for their thorough revisions. Overall, I found the manuscript to be greatly improved. Notably, I am no longer concerned about the need for English editing and the divided attention task is now well described in the methods. However, I do not feel as though my final comment concerning the hand grip task was appropriately addressed. I am unsure if this was intentional, or if the authors simply didnt understand my query. Thus, I have outlined in specific detail below:
1) Maximal grip strength - I am unsure as to why the authors revised to include post hoc testing with a non-significant ANOVA (P>0.05). I did NOT request this in my initial comments and it should be removed.
2) Strength differentiation - though I have not seen this task described in the literature before I found it a unique and insightful measure to evaluate motor control. As described in the Methods section, "the second task involved the differentiation of force by the solider in such a manner that he performed the dynamometer grip attaining half the value obtained in the first task (50% max)." Thus, performance of this task should not be evaluated by the amount of force produced, but by the ability of the solider to produce a value that is as close as possible to 50% of max. As such, performance on this measure should be expressed as a %difference from the target (i.e., 50% maximum). In this light, performance at P3 was in fact WORSE for the strength differentiation task (~20% off the target) compared to P1 and P2 where the target was only missed by ~10%. This correction should be made for both the second and third task and the manuscript should be revised throughout. A deficit in motor control following such a training exercise is logical and may have implications for solider performance.
Author Response
We are grateful for above critical remarks concerning our work. According to remark 1, post hoc testing has been removed from table 2 and from the text (lines 274-277).
As regards the strength differentiation task, our intention was to point, that no deterioration in strength occurred during as long as 48-hour survival training with sleep deprivation, since a tendency to increase maximal strength was seen after 48 h, along with higher 50%max, and high positive correlation was found between the changes in these two parameters (mentioned in our previous response). Also, analyzing real %max in this task, the results are as follows:
|
Strength max |
50%max |
Real %max |
|
|
P1 |
392,4 |
220,7 |
56,2 |
|
P2 |
367,2 |
215 |
58,6 |
|
P3 |
410,1 |
254 |
61,9 |
As seen in the table above, all 50%max exceed target 50%max. On the other hand, we have to admit, that the difference between real and target 50%max is seen at P3, which may point to some deterioration in motor control. Taking int account the reviewer comment, we have made some correction in the description of the results, as well as in their interpretation and conclusion. Also, we have added three additional references. All corrections in our revision 2 have been highlighted in yellow. Thank you for this critical remark, which allows us to realize that not only decrease but also increase in 50%max test may indicate deterioration in motor control, which undoubtedly has an application value.